# A Stress-State-Dependent Thermo-Mechanical Wear Model for Micro-Scale Contacts

**Jamal Choudhry \*** , **Roland Larsson** and **Andreas Almqvist**

Division of Machine Elements, Luleå University of Technology, 97187 Luleå, Sweden
* Correspondence: jamal.choudhry@ltu.se

**Abstract:** Wear is a complex phenomenon that depends on the properties of materials and their surfaces, as well as the operating conditions and the surrounding atmosphere. At the micro-scale, abrasive wear occurs as material removal due to plastic deformation and fracture. In the present work, it is shown that fracture is stress-state-dependent and thus should be accounted for when modelling wear. For this reason, a three-dimensional finite element model has been adopted to simulate and study the main mechanisms that lead to wear of colliding asperities for a pair of metals. The model is also fully coupled with a non-linear thermal solver to account for thermal effects such as conversion of plastic work to heat as well as thermal expansion. It is shown that both the wear and flash temperature development are dependent on the stress triaxiality and the Lode parameter.

**Keywords:** finite element method; flash temperature; wear

## 1. Introduction

Wear occurs as a consequence of two surfaces in sliding contact. The two major types of damage are abrasive and adhesive wear. Abrasive wear is the process of material removal due to high strains which occur during sliding contact, while adhesive wear refers to the material transfer resulting from the bonding between two or more solid surfaces in contact.

For ductile metals, the majority of the strain energy comes from plastic work and the large deformations can ultimately lead to mechanical failure of the interacting surface asperities. These types of mechanical failures can be seen as a form of abrasive wear and there is a great need to develop numerical models which can be used for accurate predictions. It is widely known that the mechanical failure criteria are not a constant but rather depend on the stress state of the system. Compressive strains tend to give rise to larger deformations as compared to shear and tensile strains, since the latter tend to reach the failure criteria quicker. This implies that abrasive wear in a sliding system is strongly dependent on the stress state which in turn can be dependent on other parameters such as surface roughness, material properties and thermal response.

One of the most common methods to predict abrasive wear is by using Archard's wear law [1], which states that the wear is proportional to the contact pressure. This assumption requires the knowledge of the constant of proportionality, i.e., the wear coefficient, which is difficult to quantify because it depends on many factors. It may, e.g., depend on the contact pressures as well as the stress states of the surfaces. For this reason, there is a need to develop a wear model that can account for various stress states of surfaces.

Many researchers have used the Boundary Element Method (BEM) to calculate the elasto-plastic response of rough surfaces in contact. This method provides a way to calculate the contact pressure for two rough surfaces in contact and is based on the half-space assumption. Combining BEM with Archard's wear law, it becomes possible to calculate wear based on the pressure solution [2–9]. Jacq et al. [10] and Sainsot et al. [11] used a modified BEM approach to solve the elasto-plastic contact mechanics problem. An FFT-based

contact mechanics solver was developed by Wang et al. [12] and validated by the Finite Element Method. Sahlin et al. [13] and Almqvist et al. [14] developed a contact mechanics solver which assumes that the contacting bodies are elasto-plastic and was, thereafter, verified in [15]. This model has been used in many recent investigations [16–19] and was further developed by Almqvist et al. [20,21]. It should be noted that the methods mentioned above assume that the plastic deformation is only local and not severe. The method may lead to a loss in accuracy, for example, when non-homogeneous and nonlinear problems are present. It should also be mentioned that none of the above methods can model the high-plastic strains and failure mechanisms taking place at the micro-scale contact.

In the present work, we develop an abrasive wear model from a non-linear Finite Element Method and use it to study the damage and abrasive wear of two colliding asperities. There have been many attempts to model the abrasive wear with material removal techniques such as Fang et al. [22], Jain et al. [23], Maekawa [24], Tian and Saka [25], Schermann et al. [26], Mamalis et al. [27], Liu and Proudhon [28] and Woldman [29]. In the work of [29], the abrasive wear was modelled with strain failure criteria and coupled to damage using the Johnson and Cook failure model [30]. This failure model, however, is only suitable for moderate-to-high stress triaxialities and is unable to accurately describe the failure behavior of the material at low-stress triaxialities (i.e., shearing) [31], which is what is expected for asperity contacts. Based on the mentioned previous work, it is clear that there is a need to develop an accurate three-dimensional micro-scale contact mechanics model which takes various stress states into account for the prediction of wear. High stresses and loads will cause the interacting surfaces to undergo large deformation and thus the mechanical behaviour of the material will strongly influence wear. At its core, abrasive wear is the result of mechanical failure due to high strains and there is a need to study the correlation between mechanical properties, such as flow curve, failure strains and surface roughness and wear.

The large amount of plastic work can also lead to the generation of heat within the asperities and can play a major role in the development of so-called *flash temperatures*. The flash temperatures are typically attributed to sliding frictional energy and are the main source for heat generation at the contact interface. Most of the studies that have been performed on flash temperature are based on the micro-scale and on the Carslaw and Jaeger [32,33] model of moving heat sources to describe local asperity temperature rise in semi-infinite and insulated bodies. The Carslaw and Jaeger model has been used by many researchers such as Gao et al. [34], Wang [35] Borut et al. [36] and Waddad et al. [37,38] to study transient thermal effects for various types of contact problems. The main disadvantage is that the method assumes semi-infinite and insulated bodies which may not satisfy real contact conditions, as shown by Zhang et al. [39]. While studying flash temperatures, Kennedy [40] found that more than 90% of the frictional energy is dissipated as plastic work and transformed into heat. This may imply that most of the frictional energy being converted to heat actually comes from the plastic deformation of the asperities.

In this paper, we present a numerical model for predicting the elasto-plastic and thermal response of sliding rough surfaces using the Finite Element Method and continuum damage mechanics. The numerical model includes the state-of-art failure model known as the Generalized incremental stress-state-dependent damage model (GISSMO) [41,42], which presents a phenomenological failure model formulation that allows for an incremental description of damage accumulation, including material softening and failure at different temperatures and strain rates. One of the major advantages with GISSMO is its ability to use an arbitrary definition of the triaxiality-dependent failure strain as input, which can be used to predict failure for a wide range of different stress triaxialities and materials. This makes GISSMO particularly suitable for modelling abrasive wear in contact mechanics. In order to predict the fracture as accurately as possible, the state-of-the-art fracture Modified Mohr–Coulomb (MMC) model is calibrated and used as input to GISSMO. In a comparative study conducted by Yanshal et al. [43], data from 15 different experiments were used to calibrate nine different fracture models. All models describe the failure strain

variations for different stress states. The MMC model predictions were shown to produce the least error compared to experimental values. A conclusion made in the study is that the MMC model provides the best predictability for a wide range of stress states, which underscores the motivation for using it in our study.

Our wear model is fully coupled with a thermal solver to consider the thermal effects by plastic work as well as thermal expansion. The main novelty in our work is the use of a stress-state-dependent damage model which takes triaxiality, Lode angle and material softening into account when predicting wear. This type of composite model is of utmost importance because it allows us to study and gain a deeper understanding of the wear processes at the micro-scale. The model can be used to accurately calibrate many existing simple wear laws such as, for instance, Archard's wear law. In a realistic contact, the wear rate may not be constant and depend on many parameters such as average roughness height, temperature, material properties, material non-linearity, wear/particle debris, sliding speed, deformation rate, stress state, etc. The main advantage of the present model is that it can account for all these mentioned complexities and can be used to accurately calibrate much simpler wear laws.

In the present work, we start by first considering a collision between two smooth asperities and then later adding a smaller scale secondary surface roughness to the asperities. The secondary roughness is included to study the effect of surface roughness on the otherwise smooth asperities. This approach is interesting because it allows us to study how the introduction of secondary roughness influences the deformation and wear of the asperities. The surface topographies used for the secondary surface roughness are obtained with the method developed by Pérez-Ràfols and Almqvist [44] which generates random surfaces based on a given height probability distribution and power spectrum. Because the focus of the present work is to model the fracture mechanism and material behavior of colliding asperities, only abrasive wear is considered.

## 2. Theoretical Background

### 2.1. Linear Elasto-Dynamics

Consider a linear-elastic body with density $\rho$, body forces $b_i$, tractions $t_i$ on the boundary $\mathbb{T}_t$ and fixed displacement $u_i$ on the boundary $\mathbb{T}_u$. The equation of motion for the elastic body (without viscous damping) is written as:

$$\sum_{j=1}^{3} \frac{\partial \sigma_{ij}}{\partial x_j} + \rho b_i = \rho \ddot{u}_i, \; \boldsymbol{x} \in \Omega, i = 1, 2, 3$$

$$\sigma_{ij} n_j = t_i, \; \boldsymbol{x} \in \mathbb{T}_t$$
$$u_i = 0, \; \boldsymbol{x} \in \mathbb{T}_u \tag{1}$$
$$\epsilon_{ij} = \frac{1}{2} \left( \frac{\partial u_i}{\partial x_i} + \frac{\partial u_j}{\partial x_j} \right),$$
$$\sigma_{ij} = \boldsymbol{D} \epsilon_{ij},$$

where $\boldsymbol{D}$ is the constitutive matrix relating the Cauchy stresses $\sigma_{ij}$ and strains $\epsilon$ according to Hooke's law and $n_j$ is the unit normal vector. The equations of motion can be discretized and solved with the Finite Element method. If contact between several entities is occurring, then it is necessary to impose contact constrains in the Finite Element formulation. These equations are generally used to solve for the perfectly elastic behavior of materials. Any non-linearities in the material must be accounted for by comparing the von Mises stress with the yield stress and performing the necessary updates at each load increment.

### 2.2. Modified Mohr–Coulomb

Originally proposed by Bai and Wierzbicki [45], the Modified Mohr–Coulomb (MMC) fracture model was developed to predict the strain failure of ductile fractures. It takes the triaxiality $\eta$ and Lode parameter $\bar{\theta}$ of the stress into account for the estimation of the failure strain $\bar{\epsilon_f}$:

$$\overline{\epsilon_f}(\eta,\overline{\theta}) = \left\{ \frac{K}{C_1} \left[ C_2 + \frac{\sqrt{3}}{2-\sqrt{3}}(1-C_2)\left( sec\left(\frac{\pi\overline{\theta}}{6}\right) - 1 \right) \right] \left[ \sqrt{\frac{1+C_3^2}{3}} cos\left(\frac{\pi\overline{\theta}}{6}\right) + C_3\left( \eta + \frac{1}{3}sin\left(\frac{\pi\overline{\theta}}{6}\right) \right) \right] \right\}^{-\frac{1}{n}}, \quad (2)$$

where $K$ and $n$ are the curve-fitted coefficients in the Ludwik/Holloman strain-hardening power law:

$$\overline{\sigma} = K\epsilon_p^n + \sigma_Y. \tag{3}$$

It is known that the fracture strain is not constant and can change dramatically depending on how the load is applied on the material. The MMC model, for this reason, provides a way to make the fracture criteria stress-state-dependent by taking the triaxiality factor and Lode parameter into account.

The parameters $C_1, C_2$ and $C_3$ are to be optimized based on experimental measured failure strains for different values of the triaxialities and Lode parameters. The optimization problem consists of finding $C_1, C_2, C_3$ such that the sum of residuals between the experimental failure strains and MMC predictions is minimized, i.e.,:

$$\min_{C} f(C_1, C_2, C_3). \tag{4}$$

The objective function $f$ is defined as:

$$f(C_1, C_2, C_3) = \sum_{i=1}^{m} \left( \epsilon_f^i - \overline{\epsilon_f}(C,\overline{\theta}_i,\eta_i) \right)^2 = r^T r, \tag{5}$$

where $\epsilon_f^i$ are the experimentally measured failure strains at different stress states, $m$ is the total number of experiments and $r \in \mathbb{R}^{m \times 1}$ is the residual vector. An example MMC surface calibrated for five different stress states is shown in Figure 1.

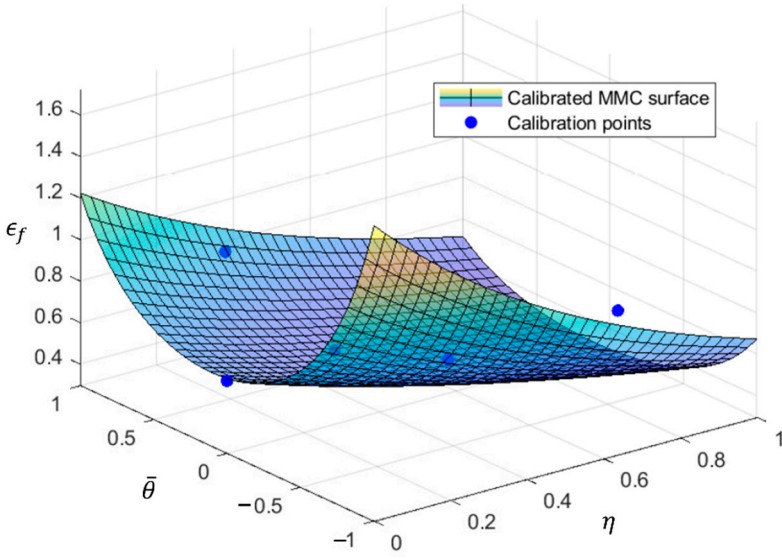

**Figure 1.** An example of the MMC surface calibrated for five different experiments [45].

The triaxiality $\eta$ is a measure of stress state and defined as $\eta = \sigma_H / \sigma_{vm}$. For $\eta = 0$ the specimen is under shear stress, for $\eta = 1/3$ it is under uniaxial tension and for $\eta = 2$ it is under biaxial tension. If $\eta$ is less than zero, the specimen is under pressure [45]. The Lode parameter ranges between $-1$ and $1$ and depends on the Lode angle. The Lode angle is also a measure of stress state and is defined as the rotation angle of the deviatoric plane around the hydrostatic axis, as shown in Figure 2.

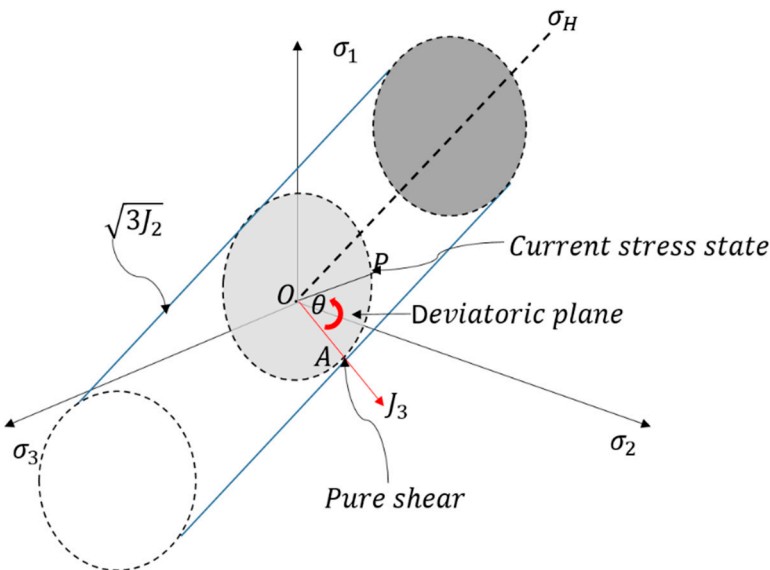

**Figure 2.** Illustration of the Lode angle in the principal stress space.

Often, the Lode angle is normalized and expressed as the Lode parameter (ranges between −1 and 1). If Lode parameter is 0, the material is under compression while if the Lode parameter is 1, it is under pure shear. As mentioned previously, the calibration of the MMC for a material is based on tensile tests which measure the fracture strain for different tensile specimens. Each specimen has its own triaxiality and Lode parameter values fixed and several specimens must be used to cover a wide range of triaxiality and Lode parameters. Figure 3 shows three examples of tensile specimens which can be used to experimentally find fracture strains for different values of triaxiality and Lode parameters.



**Figure 3.** The figure shows the dog-bone specimen (**left**) for tension, simple shear specimen (**middle**) for shear and lastly the notched specimen (**right**) for combination of shear and tension.

### 2.3. GISSMO—Generalised Incremental Stress-State-Dependent Damage Model

The idea behind GISSMO is to allow accurate modelling of the crack initiation and propagation [42]. The model is used for both ductile and brittle material failure and can accurately capture the stress evolution until fracture. The GISSMO failure model uses experimental values of fracture strains for different triaxialities and Lode angles as input and can thus be directly used together with the MMC model. The damage variable is calculated as following:

$$D = \left(\frac{\epsilon_p}{\epsilon_f}\right)^N, \qquad (6)$$

where $\epsilon_p$ is the plastic strain, $\epsilon_f$ is the fracture strain and $N$ is a calibrated damage exponent. The damage variable varies between 0 and 1 and its value is calculated incrementally:

$$\Delta D = \frac{N}{\epsilon_f(\eta,\theta)} D^{1-\frac{1}{N}} \Delta\epsilon_p. \tag{7}$$

Fracture occurs for $D = 1$. Although the damage variable is calculated at all plastic strains, the coupling to stress can be specified after a certain threshold $D_{crit}$. The effective stress is then scaled back as follows:

$$\sigma_{eff} = \sigma\left(1 - \left(\frac{D - D_{crit}}{1 - D_{crit}}\right)^M\right), \tag{8}$$

where $M$ is the calibrated fade exponent. The parameters $D_{crit}$, $M$ and $N$ determine the post-necking behaviour of the material. Depending on the value of these parameters, the material failure can be modelled as ductile, brittle or a combination of both. Figure 4 shows an example of a flow curve with both coupled and un-coupled damage.

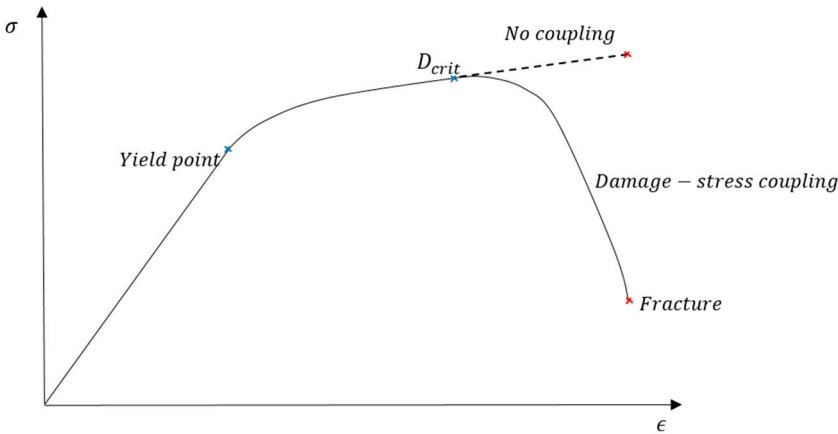

**Figure 4.** When the damage is coupled to stress at $D = D_{crit}$, the stresses are scaled back according to Equation (8) until D = 1 and fracture occurs.

### 3. Methodology

The general methodology consists of firstly generating surface topographies and then later generating the relevant Finite Element mesh. The second step would be to use mesh along with all other material and fracture data as input to the contact problem in the multi-physics non-linear FEM software LS-DYNA. An example of the contact problem can be seen in Figure 5.

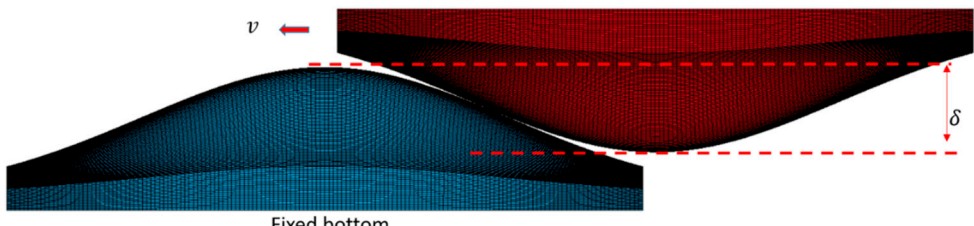

**Figure 5.** Side view of the simulation set-up. The lower body (blue) has fixed bottom nodes while the upper body (red) has a prescribed velocity in the sliding direction. The two bodies are vertically separated by an interference δ.

The contact problem is generalized to have different interferences $\delta$ as well as sliding velocities $v$. The FE mesh surfaces of the asperities shown in Figure 5 are generated from a smooth Gaussian surface with no additional micro-scaled *secondary roughness*. As will be

shown later, in the present method we will also generate and add a smaller scale *secondary roughness* on the surface of the smooth asperities to study its effect on wear development.

### 3.1. Surface Topographies

　　As mentioned in the previous section, the secondary surface roughness topographies are generated by the method developed in [44] and consist of one single surface topography in the micro-meter scale that are scaled to have different average roughness heights $S_a$. The secondary roughness was then added to the geometry of the Gaussian asperity as shown in Figure 6. The main reason for adding the secondary roughness to the asperities was to study its influence on the wear and temperature development of the sliding system since in real surfaces, asperities are rarely perfectly smooth. The area of the surfaces with the secondary roughness was 0.015 mm × 0.015 mm and divided into 256 × 256 grid points. After the surfaces were generated in MATLAB, they were used to construct the final mesh used in the FE model. This was achieved by first converting the surfaces to STL format and loading in the CAD software NX Siemens, after which it is was possible to directly generate the FE mesh of the surfaces in the CAD software. Tables 1 and 2 summarize the roughness values of the topographies. Figure 7 shows the final topographies as viewed from above.

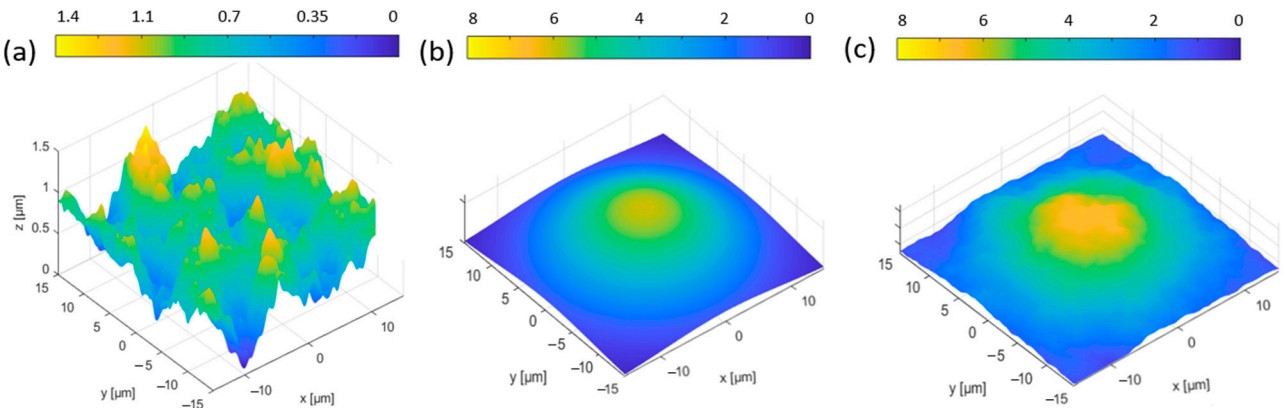

**Figure 6.** The figure shows the secondary roughness surface topography with an $S_a$ value of 0.75 μm in (**a**), the smooth asperity in (**b**) and the final surface in (**c**) obtained after the surface in (**a**) is added to the surface in (**b**).

**Table 1.** The different surface roughness topographies used in the study. All topographies had a high frequency cut-off of 0.03 and Hurst exponent of 0.8.

| Secondary Roughness | Skewness | Average Roughness Height [μm] | RMS Height [μm] |
| :---: | :---: | :---: | :---: |
| i | −0.006 | 0.75 | 0.78 |
| ii | −0.006 | 1.5 | 1.564 |

**Table 2.** The different surface topographies used in the study and their corresponding roughness values.

| Surface # | Secondary Roughness | Skewness | Average Roughness Height [μm] | RMS Height [μm] |
| :---: | :---: | :---: | :---: | :---: |
| 1 | - | 0.55 | 2.6 | 2.68 |
| 2 | i | 0.5 | 3.38 | 3.5 |
| 3 | ii | 0.41 | 4.1 | 4.30 |

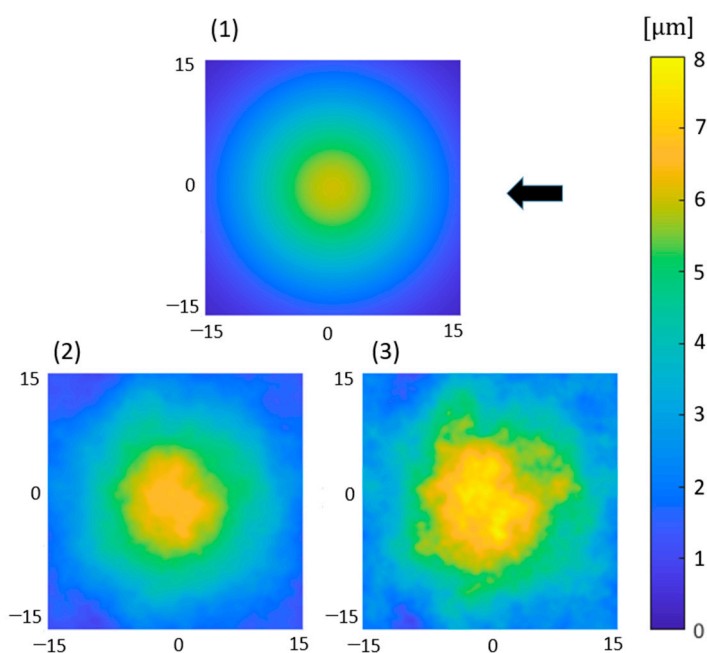

**Figure 7.** Top view of the final surface topographies used for the numerical experiments. Units are in μm and the arrow shows the sliding direction.

### 3.2. The Numerical Model

The simulation set-up consists of one upper body sliding over the stationary lower body. The bottom surface of the lower body is fixed, meaning it is constrained in all six degrees of freedom. The upper body has a fixed prescribed velocity and interference $\delta$, i.e., vertical separation, which is maintained during the entire simulation. For the contact definition between the bodies, an eroding single surface card with a pinball segment-based formulation was used. This option allows for the contact to be treated between each segment, i.e., the outer/exposed surface of each element, and allows for more accurate contact modelling as compared to a node to segment-based contact. The eroding contact card takes eroding (deleted) elements into account during the contact-searching algorithm. During the simulations, it was noted that elements that undergo large shear and compressive deformations tend to become heavily distorted and even lead to negative Jacobian volumes in the stiffness matrix. One way to solve this was to define an interior contact within the bodies, which allowed the interior elements to deform more uniformly and thus lead to less distortion. A second effective method was to improve the initial mesh quality and by using hexahedral 8-noded elements with aspect ratios close to 1. In order to improve numerical efficiency, isoparametric element formulation with single-integration Gauss point was used (ELFORM = −1). The zero-energy hourglass modes associated with the single-integration points and rank-deficiency were removed by using the Flanagan–Belytschko stiffness form hourglass control. In order to model material failure, the GISSMO damage model is implemented and element erosion (deletion) is used to simulate failure. After performing a careful mesh convergence study, the characteristic element length was set to 0.2 μm and each body consisted of approximately 400,000 elements. This ensured that the simulations were not affected by the mesh size. The numerical structural model was fully coupled with a thermal solver. An explicit integration scheme was used for the structural problem with the time step scaled to 60% of the critical time step to achieve better numerical stability.

The thermal problem implies prediction of the temperature change of the bodies due to the conversion of mechanical work to heat. For the thermal problem, an implicit Crank–Nicholson scheme was used with the maximum time step limited to 100 times the explicit structural time step. The reason for this is that the thermal process for this problem is considered much slower when compared to the structural deformations. The initial

temperature was set to 20-degrees Celsius and the bodies were thermally insulated for simplicity. An elastic and visco-plastic material model with von Mises yield criterion (material card *MAT_106) [46] and thermal expansion was used and included in the numerical model. The total simulated time was 40 μs, which was enough time to ensure that the upper body completely slides over the lower body. The study was split into two different cases. In the first case, the upper body was not allowed to deform and was modelled as rigid. In the second case, both bodies were allowed to deform. The first case represents contact between a hard and soft material and in the second case, contact between two soft materials. This approach was interesting because it allowed us to better understand the contact behavior for materials with different hardness as well as more clearly differentiate the effect of secondary surface roughness introduced in the smooth asperities.

### 3.2.1. Case 1—Rigid Body vs. Elasto-Plastic Body

In Case 1, the upper body is kept rigid (i.e., not deformable) and the lower body is deformable. The upper body has a prescribed motion while the lower body is stationary. There were a total of 3 different simulations performed in which each time a new surface (with different secondary roughness) was used for the upper body (while keeping the same surface for the lower body). Table 3 summarizes the parameter values used for Case 1.

**Table 3.** Parameter values for the simulations in Case 1.

| Simulation # | Upper and Lower Surfaces | Interference $\delta$ [μm] | Speed $v$ [m/s] |
|---|---|---|---|
| A | 1 & 1 | 4 | 1 |
| B | 2 & 1 | 4 | 1 |
| C | 3 & 1 | 4 | 1 |

### 3.2.2. Case 2—Elasto-Plastic Body vs. Elasto-Plastic Body

In Case 2, both bodies are deformable and the same mesh, sliding speed and interference was used as with Case 1. The lower body is stationary while the upper body has a prescribed motion. Table 4 summarizes the parameter values used for Case 2.

**Table 4.** Parameter values for the simulations in Case 2.

| Simulation # | Upper and Lower Surfaces | Interference $\delta$ [μm] | Speed $v$ [m/s] |
|---|---|---|---|
| D | 1 & 1 | 4 | 1 |
| E | 2 & 1 | 4 | 1 |
| F | 3 & 1 | 4 | 1 |

### 3.2.3. Material Properties and Damage Parameters

The material properties used for this study are for a high-strength TRIP (TRansformation-Induced-Plasticity) steel and its stress–strain relationship at two different temperatures are obtained from the work of Wolfgang et al. [47], as shown in Figure 8. For intermediate temperature values in the simulation, linear interpolation and extrapolation are used to approximate the correct stress–strain relationship. The effective plastic strain refers to the true strain after elastic unloading and the stress is the true stress. All material properties are shown in Table 5. For simplicity, the same material properties and damage parameters were used for all bodies.

**Table 5.** Material properties.

| Elastic Modulus [GPa] | Poisson Ratio | Density [kg/m$^3$] | Specific Heat Capacity [J/(kg K)] | Thermal Conductivity [W/(mK)] | Thermal Expansion Coeff. [1/K] |
|---|---|---|---|---|---|
| 210 | 0.3 | 7850 | 480 | 50 | $0.12\times10{-}4$ |

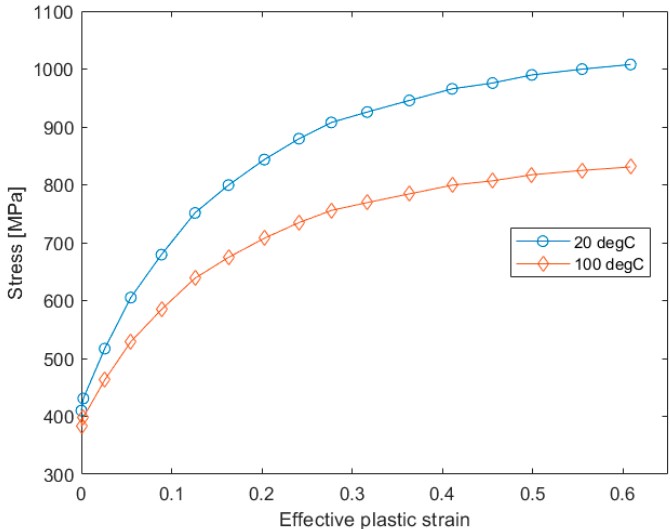

**Figure 8.** Stress–strain relationship for two different temperatures adapted from [47].

In order to increase the explicit time step size and thus obtain faster simulation results, the mass of the asperities were scaled by 1000. The artificial mass scaling did not significantly affect the wear and thermal results and was considered to be adequate for this study. As will be shown in the next section, energy data from the impact show that the kinetic energy is very small as compared to the total internal energy and thus the mass bears very little significance for the problem. For simplicity, the contact between the bodies was considered to be frictionless, and the frictional force was assumed to arise from the collision force between the asperities. The fracture characteristics for the high-strength TRIP steel were obtained by Bai and Wierzbicki [45] and are shown in Table 6.

**Table 6.** Fracture tensile tests obtained from [45].

| Test No. | Test Specimen | $\eta$ | $\bar{\theta}$ | $\epsilon_f$ |
|---|---|---|---|---|
| 1 | Dog-bone tension | 0.379 | 1 | 0.751 |
| 2 | Flat specimen with notch | 0.472 | 0.496 | 0.394 |
| 3 | Biaxial tension | 0.667 | −0.921 | 0.950 |
| 4 | Butterfly tension | 0.577 | 0 | 0.460 |
| 5 | Butterfly shear | 0 | 0 | 0.645 |

The calibration of the MMC, i.e., finding the MMC coefficients that minimized the residual of Equation (5), is performed using the Gauss–Newton algorithm and the results, along with the Ludwig/Holloman coefficients, are shown in Table 7.

**Table 7.** The optimized MMC coefficients.

| $n$ | 0.265 |
|---|---|
| $K$ [MPa] | 1276 |
| $C_1$ | 0.136 |
| $C_2$ [MPa] | 710 |
| $C_3$ | 1.068 |

The damage parameters should be calibrated based on stress–strain curves obtained from tensile tests and are important to correctly predict the post-necking behavior of the material. The chosen values of the damage parameters for this study are shown in Table 8.

**Table 8.** GISSMO parameters.

| Damage Exponent $N$ | Critical Damage $D_{crit}$ | Fade Exponent $M$ |
|:---:|:---:|:---:|
| 2 | 0.8 | 1 |

Finally, an exemplary 2D MMC curve relating the fracture strain to triaxiality is shown in Figure 9.

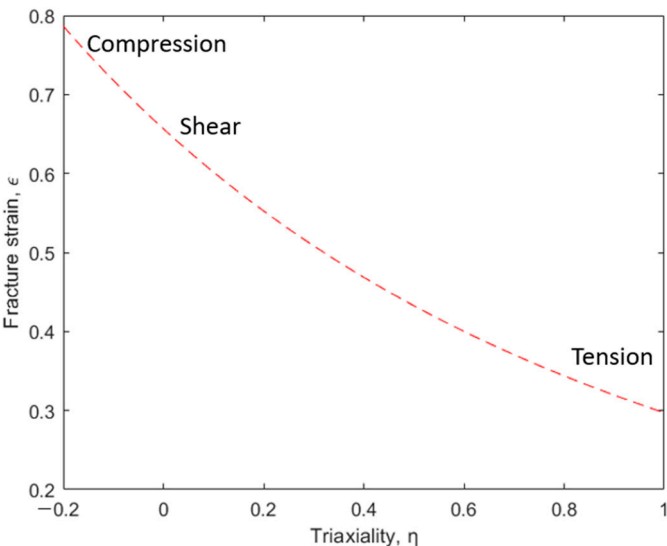

**Figure 9.** Fracture strain versus triaxiality for Lode parameter $\bar{\theta} = 0$.

To summarize, the general methodology can be described as follows:

- Generate surface topographies using the method developed in [44].
- Generate high-quality mesh for the surface topographies.
- Calibrate the MMC model for fracture strains as function of different stress states as well as damage parameters to be used as input to GISSMO.
- Set-up the contact problem in the multi-physics Finite Element software LS-DYNA.
- Run the model and analyze the wear results.

## 4. Results and Discussion

This section will present the numerical results. The results will be divided into each of the two cases as described in the previous section. All simulations were performed with the R12 version of LS-DYNA developed by Livermore Software Technology Corporation and run on 64 Xeon Gold 6248R 3 GHz cores computer cluster which took roughly 5 h in real time to solve for each simulation.

### 4.1. Model Verification for the Linear Elastic Case

In order to verify the accuracy of the model for the linear elastic case, the simulation results will be compared with those obtained by the contact mechanics solver developed by Sahlin et al. [13] and Almqvist et al. [9,14]. The simulations are set-up such that the bodies are stationary and a normal load is applied, as shown in Figure 10. The same mesh employed in Cases 1 and 2 will be used and a linear elastic material model will be applied for both bodies with the same material properties shown in Table 5.

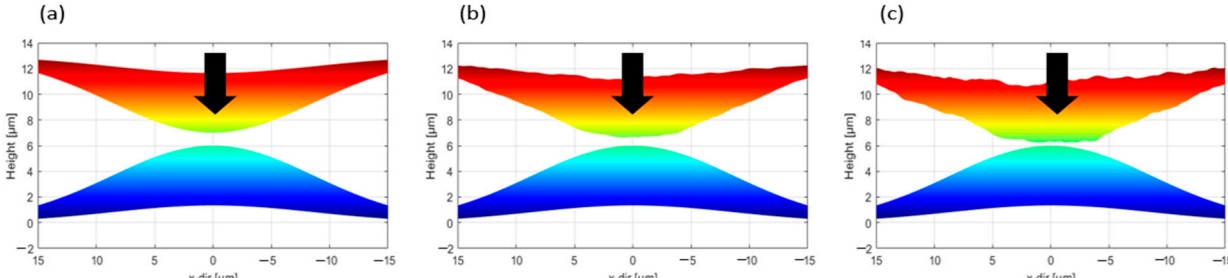

**Figure 10.** Simulations used to compare the present solution with BEM. A normal load is applied and the bodies are assumed to be linear elastic. Surface 1, 2 and 3 are used for the upper surfaces and shown in (**a**–**c**), respectively. Surface 1 is used for the lower surface for all simulations.

The maximum contact pressure and its comparison with BEM is shown in Figure 11 for different normal loads. As can be seen, there is a good agreement between the present solution and those obtained with BEM.

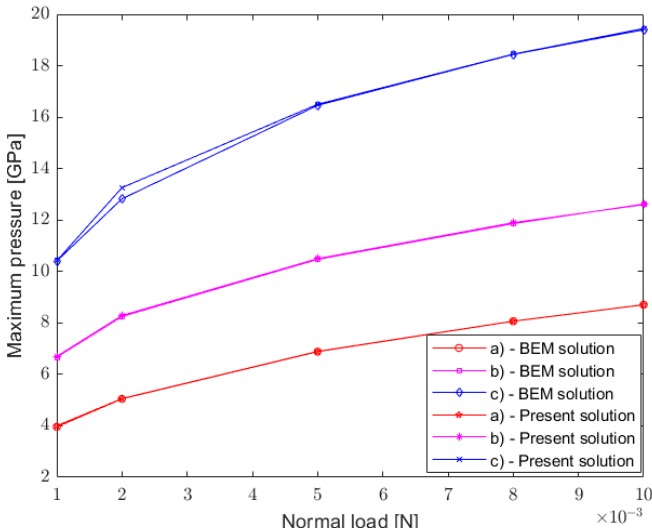

**Figure 11.** The maximum contact pressure for different normal loads and its comparison with BEM.

### 4.2. Case 1—Rigid Body vs. Elasto-Plastic Body

Figure 12 shows a snapshot of Simulation C at two different time instances. As will be shown later, the lower body experiences compression and shear at the point of impact. As the upper body moves further to the left, cracks start to form around the highly deformed elements which eventually creates a fracture path along the sliding direction. In order to further understand the impact mechanism, the strains and triaxiality results are shown for Figures 13 and 14.

As can be seen in Figure 13, areas under high compression are shown in dark blue while areas under pure shear and tension are shown in yellow and red, respectively. The area under compression is right in front of the impact zone and has large negative triaxiality values. The compression occurs mainly due to the highly deformed elements in front of the impact zone being "squeezed" as the upper body moves to the left. Because the upper body is rigid, the deformation of the elements in the lower body are much greater than if the upper body had been allowed to deform. This is because the nodes of the rigid elements are constrained to only move in the sliding direction, which forces the elasto-plastic nodes of the lower body to move with the rigid nodes. In Simulation A, the impact zone can be seen to have a large triaxiality range. This is because the pressure distribution is more uniform and there is no secondary roughness introduced in the asperities. Elements right at the end of the impact zone are being "dragged" and are thus elongated due to tension.

Immediately between the compression and tension zone lies the shear zone, which is where the triaxiality values are close to zero. While studying the results of Simulation C, one can observe that the triaxiality range is much smaller as compared to Simulation A. This is mainly due to the added secondary roughness in the upper body, which causes more shearing around and inside the impact zone and limits the triaxiality to values closer to zero. A consequence of the large negative triaxiality values of Simulation A is that the elements can undergo large strains without failing. This is because the material is more sensitive to failure for shear and tension, rather than compression. This behavior is also what the MMC surface predicts, as shown in Figure 9. Because the asperities undergo large deformations, a significant amount of plastic strain energy is converted to thermal energy. Strain results shown in Figures 13 and 14 indicate that the lower body in Simulation A deforms more than the lower body of Simulation C, which in turn also affects the temperature distribution. The strain and triaxiality results for Simulation *B* follow the same trend and those results will not be shown here. Instead, a summary will be shown for all three cases in the form of worn mass and average temperature development of the asperities as a function of time. The results are shown in Figures 15 and 16.

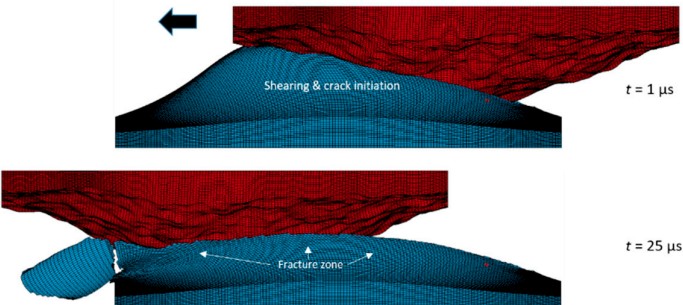

**Figure 12.** As the upper body moves, the lower body can be seen to plastically deform. The crack around the contact initializes and spreads quickly, causing abrasive wear on the lower body.

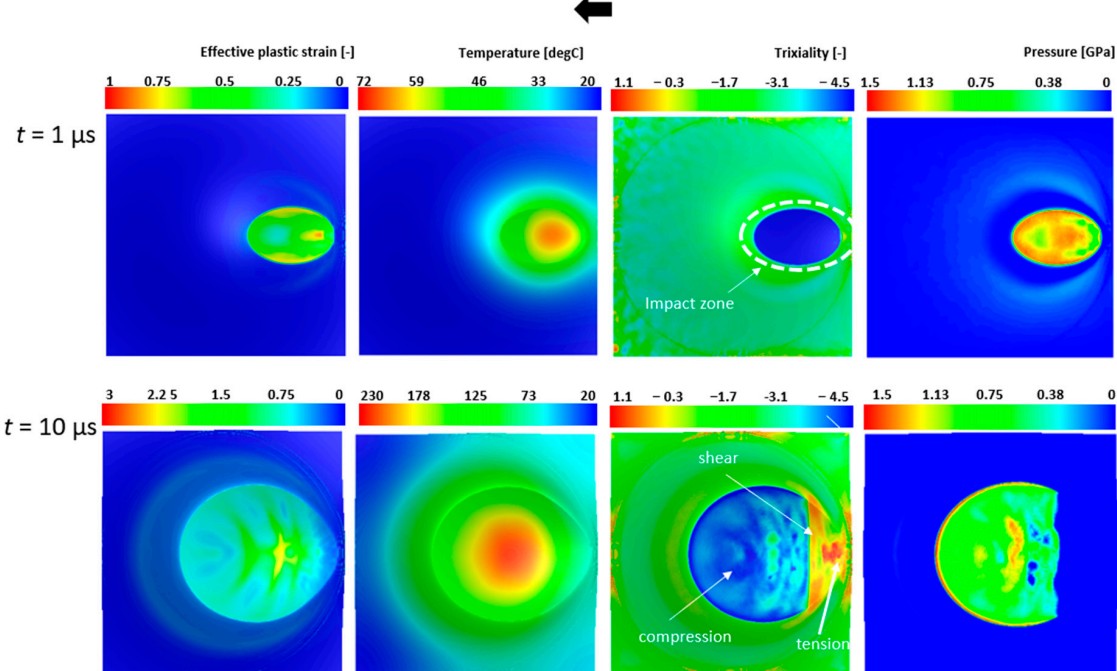

**Figure 13.** Top view of the strain, temperature, triaxiality and pressure results at two different time instances for Simulation A. The triaxiality shows all three stress states, i.e., compression (blue), shear and compression (green), pure shear (yellow) and tension (red). The arrow shows the sliding direction.

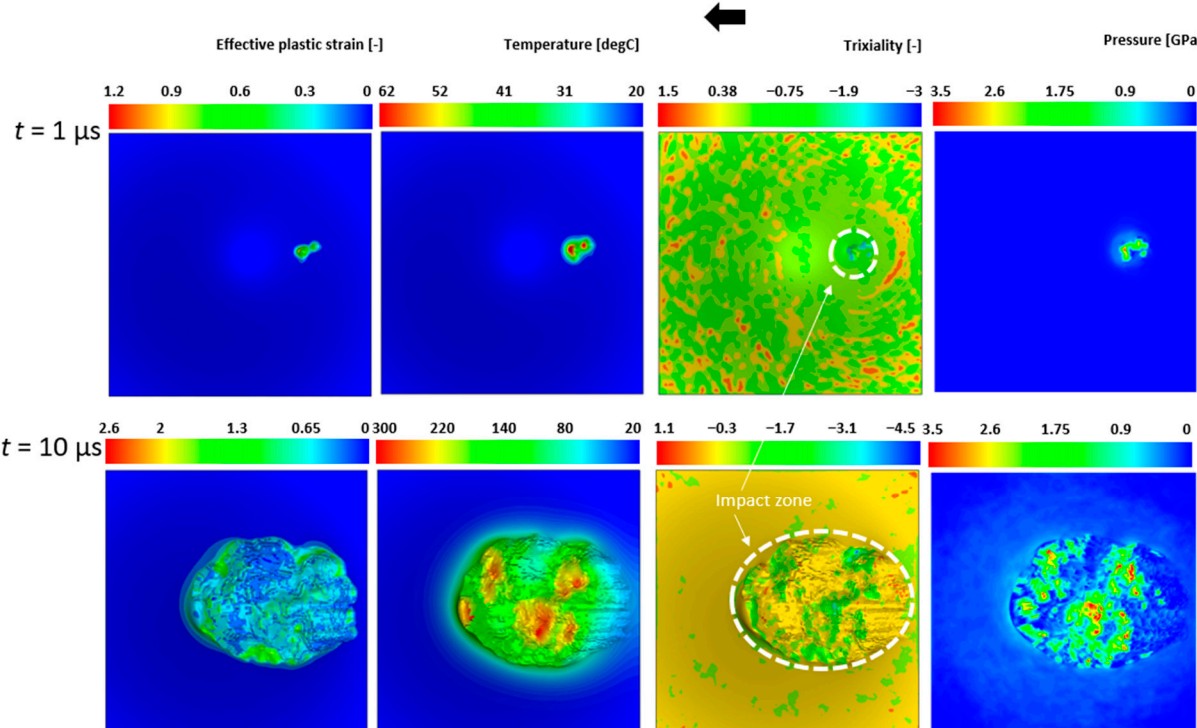

**Figure 14.** Top view of the strain, temperature, triaxiality and pressure results at two different time instances for Simulation C. The triaxiality result shows a larger dominance of shear stress around and inside the impact zone. The arrow shows the sliding direction.

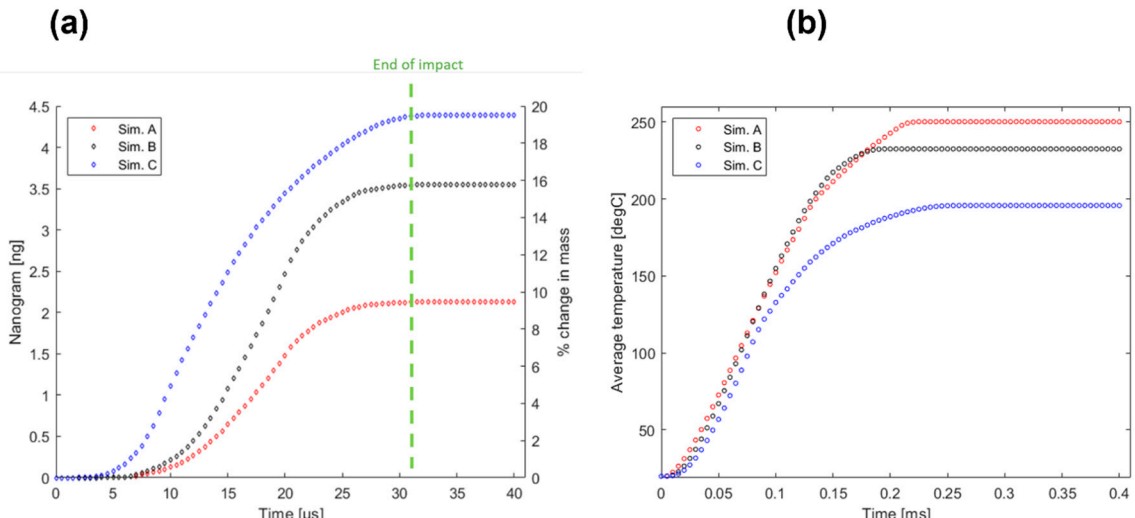

**Figure 15.** Worn mass of the lower body shown in (**a**) and average temperature in (**b**).

The wear development can be seen to exhibit an exponential growth during 0–15 µs and then a logarithmic behavior from 15–40 µs. The temperature development can be seen to exhibit a similar logarithmic increase as the wear mass increases and as the impact nears its end, the temperature reaches a steady-state value. As expected, the average temperature change of the simulations is influenced by the secondary roughness. As the $S_a$ value of the secondary roughness decreases, the temperature change increases due to less shearing and larger negative triaxiality values inside the impact zone. This indicates that the temperature development is strongly dependent on the triaxiality values. Figures 17 and 18 show the transient force development. It can be seen that the normal force is greater for Simulation

A as compared to Simulation B and C. With more secondary roughness, the decrease in the real area of contact causes a significant drop in the normal force. However, the general trend is that the maximum tangential force as well as the coefficient of friction increases with increasing secondary roughness.

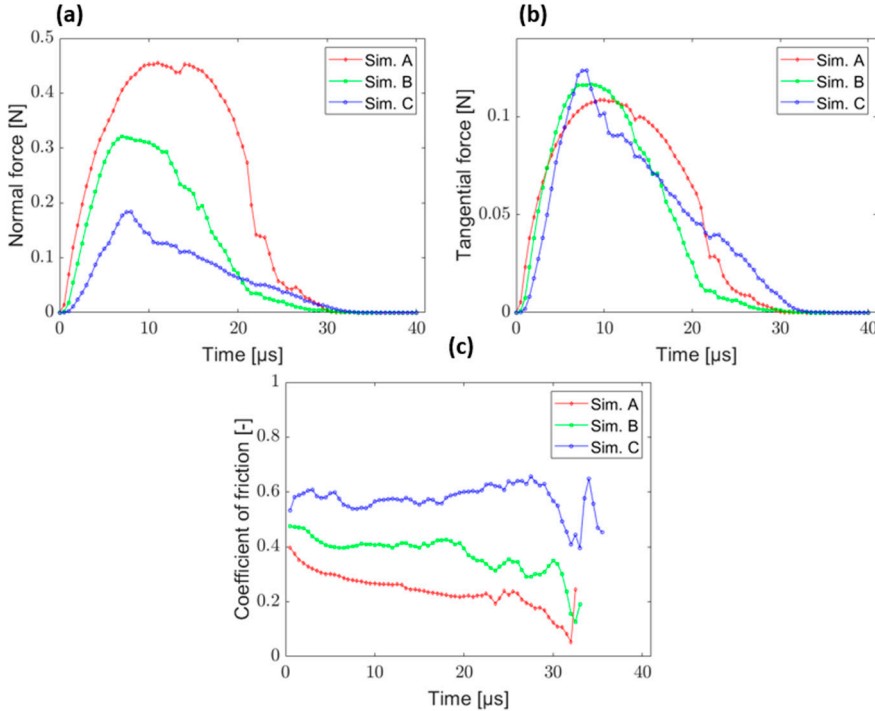

**Figure 16.** Normal force shown in (**a**), tangential force in (**b**) and the coefficient of friction in (**c**).

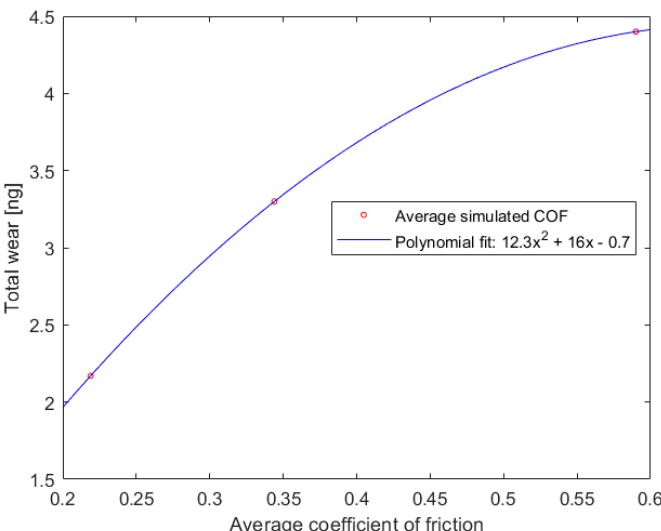

**Figure 17.** The average simulated COF together with a polynomial fit.

It can be seen that the development of the coefficient of friction behaves as expected. It is highest during the beginning of impact and gradually declines as more and more mass is worn. The average coefficient of friction during the whole impact was measured and shown in Figure 17. The results indicate that wear is directly linked to friction, as expected. The total wear at the end of the simulation increases as friction increases (due to an increase in the average roughness height). Figure 18 shows the final topography of the worn asperities with clearly visible wear tracks (due to the secondary roughness) and areas of large material removal. Figure 18 also shows the final strains for all simulations at the

end of impact and it can be clearly seen that Simulation A bears larger strains as compared to the other simulations. Again, this is a result of different stress states for each simulation and the material failure sensitivity for different triaxiality values. The wear tracks due to the secondary roughness are more prominent for Simulations B and C. Finally, a mesh convergence analysis is shown in Figure 19. No significant differences in the results were observed for element size below 0.2 μm.

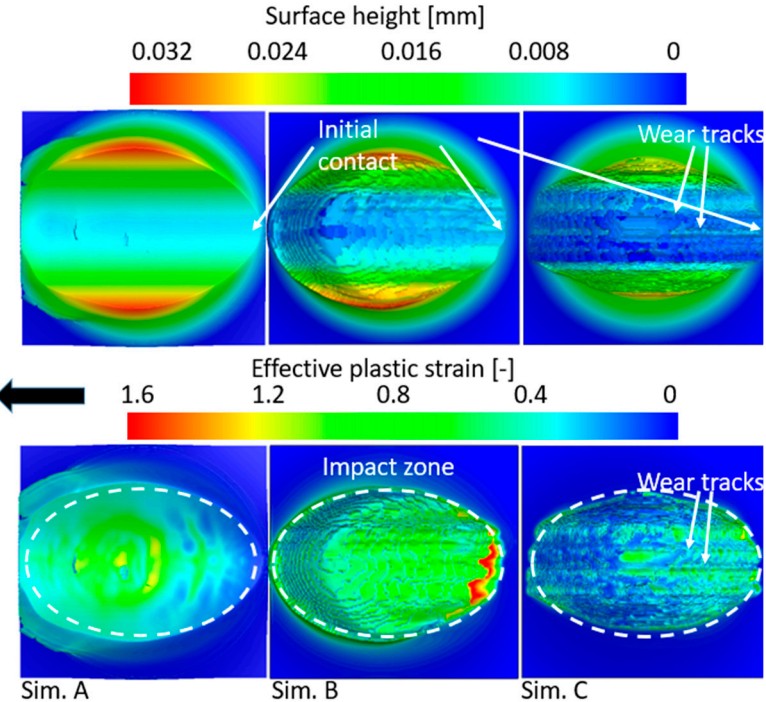

**Figure 18.** Top view of the surface height of the lower body for all three simulations at the end of impact, i.e., t = 40 μs (**top**) and the final plastic strains (**bottom**). The impact zone is confined within the dashed area and the arrow shows the sliding direction of the upper body.

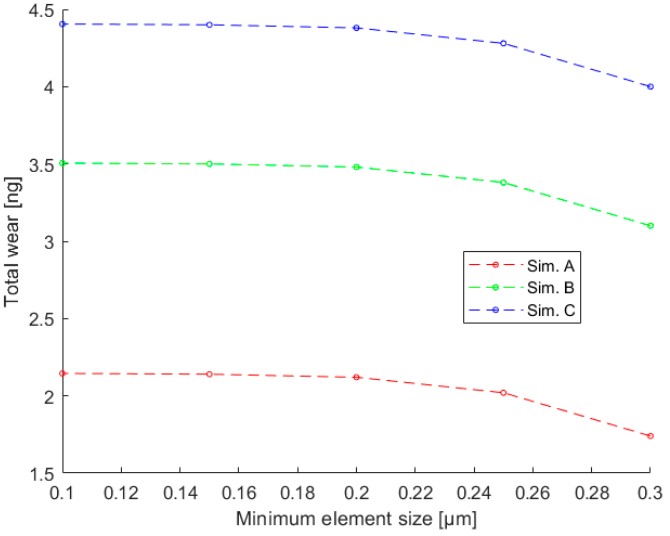

**Figure 19.** Mesh convergence analysis.

*4.3. Case 2—Elasto-Plastic Body vs. Elasto-Plastic Body*

The worn mass and average temperature development for the simulations of Case 2 are shown in Figure 20. As can be seen, the wear development here exhibits the same behavior as seen with Case 1. That is, the wear rate increases between 0–15 μs and then slowly

decreases between 15–40 µs. The same behavior is seen with the temperature development. The development of forces and the coefficient of friction are shown in Figure 21. The normal forces as well as the coefficient of friction in Case 2 follow the same trend as seen in Case 1, with the exception that now there is less difference in the maximum force between each simulation. Interesting to note is that the influence of the secondary roughness is not as clear here as compared to Case 1, i.e., the wear development looks almost the same for all simulations. The main reason for this behavior may be due to the deformability of the upper body. In Case 1, the rigid nodes allowed for high-stress concentrations to exist during the whole impact process. The same cannot be said with Case 2 as the nodes are deformable and cause the high-stress concentration zones to quickly fade. In order to display this behavior more clearly, the triaxiality result of the upper bodies is shown in Figure 22 at two different time instances.

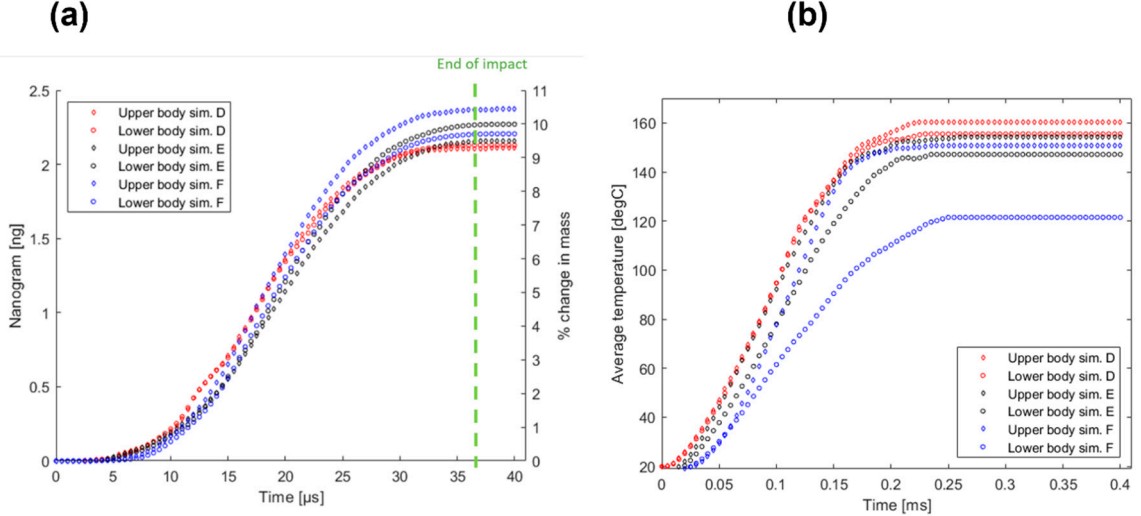

**Figure 20.** Worn mass of all bodies shown in (**a**) and average temperature in (**b**).

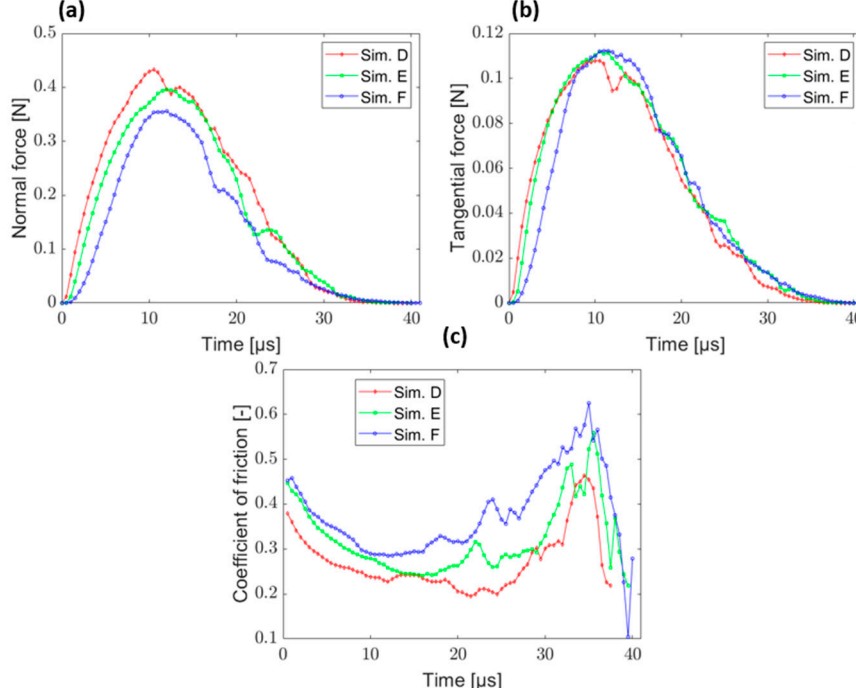

**Figure 21.** Normal force shown in (**a**), tangential force in (**b**) and the coefficient of friction in (**c**).

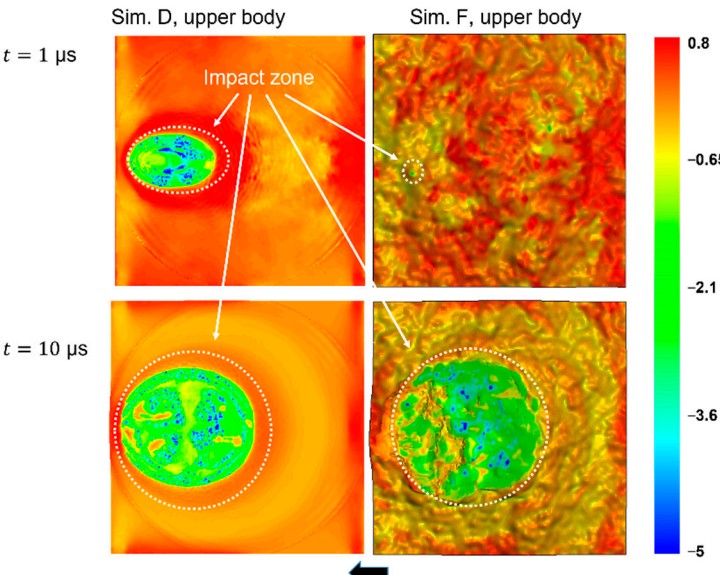

**Figure 22.** Top view of the triaxiality result for the upper bodes of Simulation D and F in Case 2. The stress states are dissimilar during initial contact as seen in the upper figures. The differences, however, fade with time as the upper body deforms and loses its sharp peaks during impact, as shown in the lower figures.

As can be seen from Figure 22, the high-stress concentration zone in the upper body of Simulation F causes significant shearing during the initial contact. For Simulation D, there is a mixture of shearing and compression. During the initial contact, there is a clear difference of stress state between these two simulations. The difference, however, quickly fades as the upper body moves and begins to deform. The high-stress concentration occurring at the sharp peaks of the upper body tends to cause extreme shearing and large plastic deformations were observed in those zones. If the peaks were large enough, the extreme shear stresses caused the peaks to wear off quickly through brittle fracture. It was noted, however, that for peaks that laid below a certain height limit, compressive stresses dominated and the peaks in those zones plateaued out through the plastic deformations. The results show that the stress state of all simulations behaves similarly after the sharp peaks (i.e., peaks with high stresses) fade resulting in similar wear and temperature development regardless of secondary roughness. The plateauing behavior is seen more clearly in Figure 23, where the stress triaxiality is also shown. Nonetheless, the wear development in Simulation F is still slightly higher than Simulation D and E. This indicates that the initial sharp peaks, which although fade quickly, may still have a small influence on the subsequent wear development during the impact. The triaxiality result for Simulation E follows the same trend and its result will not be presented here. Instead, the final worn topography and strains are shown in Figures 24 and 25.

While comparing the topography of the worn bodies, it can be noted that they are subjected to almost the same amount of wear and that there are only some minor differences between each simulation. The highest peaks, shown in orange and bright red, are the "outer" impact zones with the elements there having slightly less plastic strain as compared to the zone in the middle of the impact. Rather than being exposed to extreme shearing, the elements in the outer impact zone have been "pushed" to form ridges and are thus the highest peaks. As expected, the highest plastic strains are located just in front of the impact zone, as the elements there are the first to make initial contact and are subjected to extreme stress. Very similar results were observed with Case 1, but one of the main differences one can note is that the length of the impact zones in the sliding direction is slightly longer than Case 2. Again, this is another consequence of the deformability of the upper body in Case 2. As the upper body deforms and is worn, it loses its ability to transfer high-stress

concentration on the lower body and thus reducing its ability to deform it. This in turn results in less wear and a shorter impact zone length as compared to Case 1.

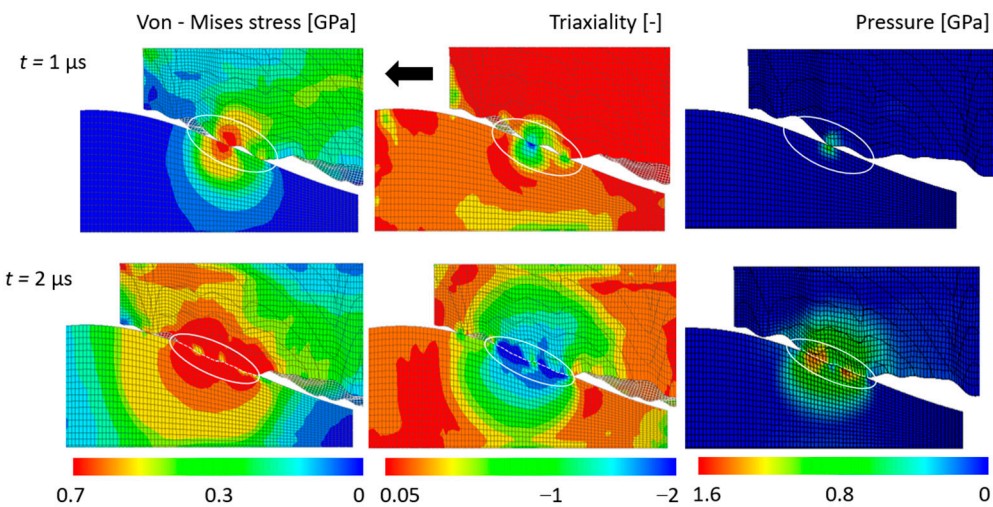

**Figure 23.** Cross-section view of the effective stress (**left**), triaxiality (**middle**) and pressure (**right**) at two different time instances for simulation F. The impact zone is confined within the white ellipse.

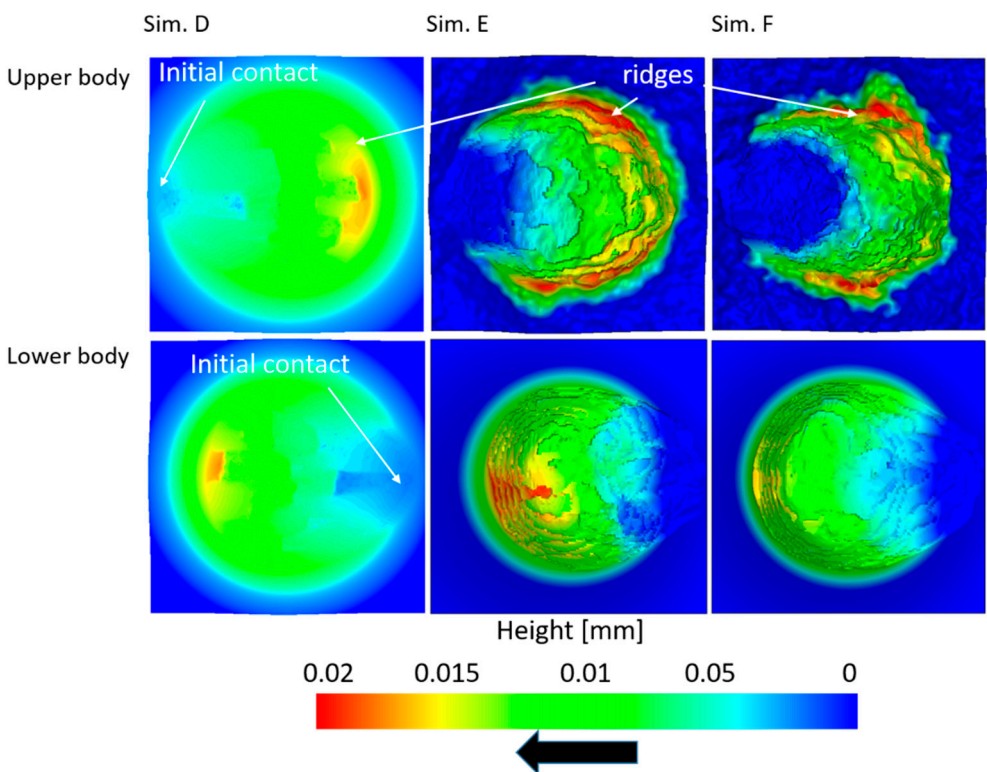

**Figure 24.** Final topographies of the bodies as viewed from the top and obtained at time 40 μs.

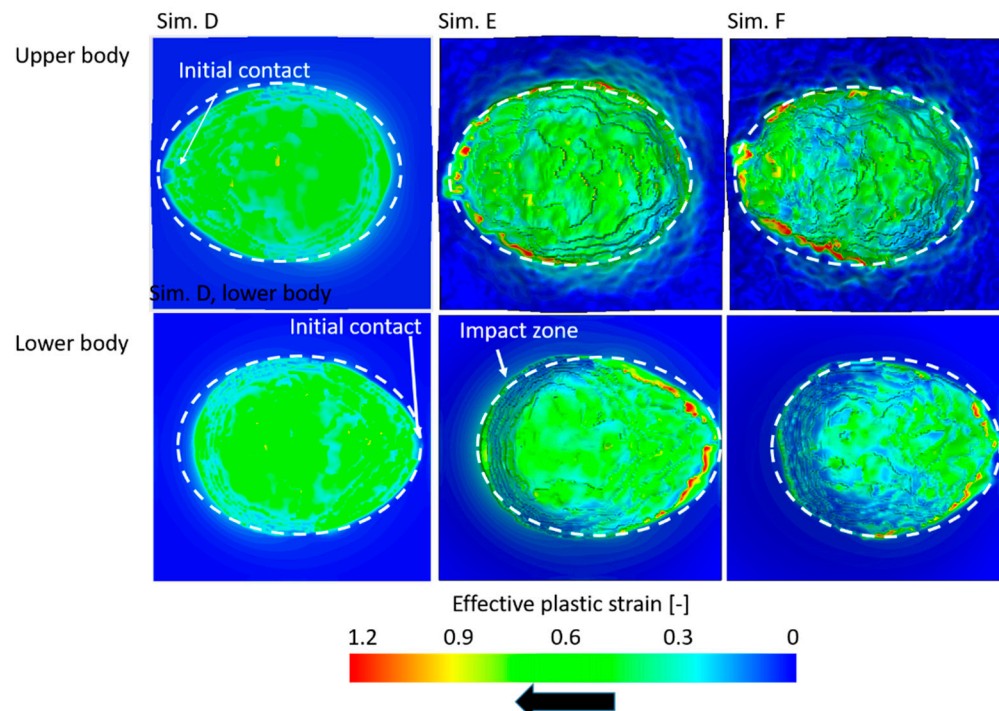

**Figure 25.** Effective plastic strain of all bodies as viewed from the top and at time 40 μs.

Because the temperature development is strongly dependent on the plastic deformations, it is clear that accurate thermal modelling relies on the GISSMO damage parameters. This is because the post-necking, such as brittle or ductile, behavior significantly affects the number of plastic deformations. For brittle fracture, there is less plastic work because the fracture is more sudden and no softening occurs, see Figure 6. This shows the importance of including damage when simulating the flash temperature development in asperities. Finally, Figure 26 shows the internal strain energy and kinetic energy. As can be seen, the kinetic energy is much lower than the strain energy even after artificial mass was introduced to increase the explicit time step size. In other words, the mass scaling does not affect the solution and is an effective way to obtain much faster simulation times.

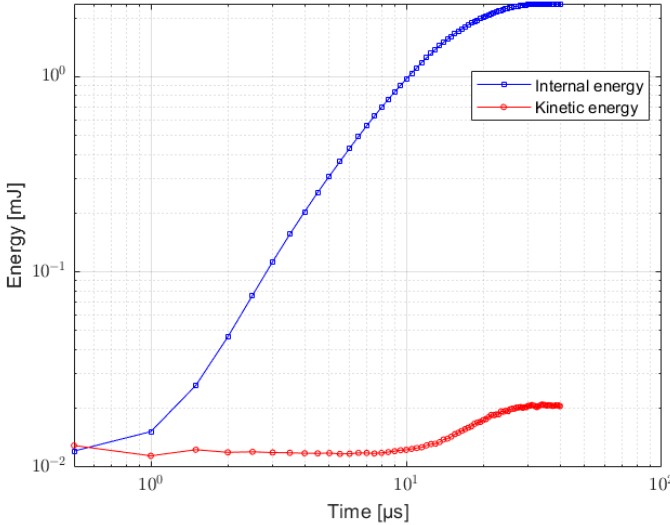

**Figure 26.** Internal strain energy and kinetic energy. The maximum kinetic energy reached is approximately 2% of the maximum reached internal energy.

### 4.4. Comparison with BEM

Because the proposed FE model is three-dimensional and based on a realistic fracture mechanics model, it may be difficult to directly compare it with other simpler wear models. It can, however, be shown that it is possible to use the simpler models by calibrating them such that they give similar results obtained with the present method. This approach can be useful for saving computational resources and time. In this section, a correlation of the wear calculations between the present solution and the BEM contact mechanics solver from [9,13,14] will be made for Case 2. In BEM, Archard's wear law can be used to compute the wear depth at each time increment [9,13,14]. The Archard's dimensionless wear coefficient was determined iteratively and chosen such that the wear results of Sim. D in the BEM calculation gave similar results obtained with present method. The coefficient $3 \times 10^{-5}$ was found to give the best match between the BEM calculation and present FE model for Sim. D, as shown in Figure 27b. The Hardness of the material was set to 1.5 GPa and all operating conditions were kept the same as with Case 2. The wear results show that for this specific wear coefficient, there is a reasonably good agreement between present work and BEM. Although Archard's wear law is typically used for determination of the global wear and is much simpler to use than the advanced proposed model, it shows that it is possible to use the proposed FE model to calibrate Archard's wear law for a specific problem. This in turn allows one to re-run the same problem in BEM and approximate wear for different geometries and surface roughness, as shown in Figure 27a for Sim. F. Worth noting, however, is that the wear coefficient is determined for a single interference in the current FE model. In reality, this value is dependent on different interference values, and it may be necessary to re-calibrate Archard's wear law for problems with different interferences simulated by the present FE model. For three-dimensional rough surfaces, the approach would be similar.

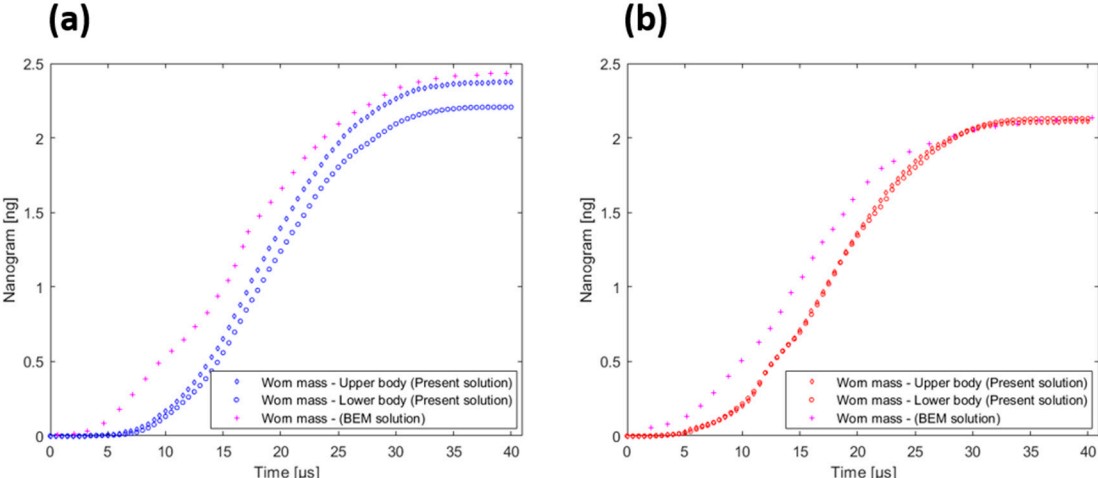

**Figure 27.** Comparison between the present solution and BEM for Sim. F shown in (**a**) and Sim. D shown in (**b**).

It would be possible to use the present model to predict wear and then re-run the same problem in BEM with Archard's wear law. The wear coefficient would then be determined such that the wear results in the BEM calculation provide similar results with the present model. Once the right coefficient is found, it would be possible to run the BEM calculations again but for different surface roughness/topographies and operating conditions.

## 5. Conclusions

A three-dimensional finite element model was introduced to solve the contact mechanics problem with permanent damage of the surfaces. The present model takes these stress states as well as thermal effects into account when predicting wear. It was shown

that the damage, i.e., the wear, is a stress-state-dependent phenomenon that depends on the triaxiality and Lode parameters. The model can be used to calibrate much simpler wear models such as Archard's wear law. It is, for instance, possible to determine the wear coefficient in Archard's law for different contact conditions and parameters and thus re-run the same problem in a simpler wear model. It is worth mentioning that, although the present solution only models abrasive wear, the model can be used to model adhesive wear as well.

To simulate adhesion, the model would then require additional "full stick" constraint in the contact algorithm. From the results presented, the following conclusions can be drawn:

- When a rigid body collides with an elasto-plastically deformable body, the secondary roughness has a significant effect on the wear and temperature development.
- When two elasto-plastically deformable bodies with the same material properties collide, the effect of the secondary roughness is significantly reduced.
- The wear development is strongly dependent on the triaxiality and Lode parameter, and compressive stresses tend to lead to less wear as compared to shear and tension.
- The flash temperature development is also dependent on the stress state, with compressive stresses leading to higher temperature increases as compared to shear and tension.

**Author Contributions:** Conceptualization, R.L., J.C. and A.A.; methodology, J.C.; software, J.C.; validation, J.C.; formal analysis, J.C.; investigation, J.C.; resources, R.L. and A.A.; data curation, J.C.; writing—original draft preparation, J.C.; writing—review and editing, R.L. and A.A.; visualization, J.C.; supervision, R.L. and A.A.; project administration, R.L.; funding acquisition, R.L. All authors have read and agreed to the published version of the manuscript.

**Funding:** This research was funded by the Swedish Research Council with grant number 2020-03635.

**Conflicts of Interest:** The authors declare that they have no known competing financial interests or personal relationships that could have appeared to influence the work reported in this paper.

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
