# Peer review of "A Stress-State-Dependent Thermo-Mechanical Wear Model for Micro-Scale Contacts"

_lubricants, doi:10.3390/lubricants10090223_

Round 1

Reviewer 1 Report

In this manuscript, the authors developed a stress-state dependent thermo-mechanical wear model for micro-scale contacts. They constructed a three dimensional analysis with considering various effects such as the roughness, temperature, and material properties, velocity, and simulated the behaviors in different cases. The manuscript is well-organized, the present model works well, and some novel findings are obtained. Therefore, I believe the paper is suitable for publishing on Lubricants after minor revision. My questions are listed as follows:

1. Although the data are abundant, there are too many individual figures in the manuscript, making the manuscript too lengthy. Many of the figures could be combined, e.g., for Case1 Figures 15-16; Figures 17-19.

2. In Figure 20, is there any reason for using a polynomial fit instead of linear fit?

3. In Figure 23, only the analysis of Sim. A is shown. Could the authors please compare that with Sims. B&C?

4. It is better to briefly define the “secondary roughness” when it first appears with 1-2 sentences to avoid any misleading, as this word is quite important in this manuscript.

5. Figures 6a & 21 and etc., the values on the bar are not in an arithmetic sequence, please check that.

6. Please check the figures carefully. On the left of Figure 33, there are some weird sentences covered by the figure. There are also something similar in other figures.

7. The references are quite confusing, and I suggest the authors to formalize the references. It took me some time to realize that Ref. 16 and Ref. 22 are referring to the same paper.

Reviewer 2 Report

As per my observation author should highlight the novelty of this work as well as application of the model presented.

Reviewer 3 Report

The authors present a finite element-based numerical method for the simulation of the wear process of a single pair of asperities during one passage. While the study is interesting and relevant to the field of tribology there are several concerns that should be addressed before any further steps. 

1.       Abstract: it should be stated that the model is developed and used to simulate a pair of metals (steels). If polymers or composites would be studied, most likely, different types of material models, e.g. considering viscoelastic properties, would need to be used.

2.       As stated by the authors, wear is formed by adhesive and abrasive types. The abrasive wear is investigated in the study, however, it should be clarified if and when abrasive wear can be decoupled from the adhesive component in numerical models, i.e., if the adhesive component doesn’t play a role in the abrasive wear simulation as described in subsequent pages. If that is indeed the case please provide suitable references for that

3.       The flash temperature models as presented in https://doi.org/10.1115/1.2927035 and  https://doi.org/10.1016/j.jcde.2019.03.001 also follow the same theory and are applied to more complex tribological systems.

4.       Equations: Descriptive indices that don't represent a numeric index are typically written in regular text, i.e., non-italic.

5.       3. methodology – use capitalization

6.       Sect. 3.1 – for the surface generation in Matlab, have any specific libraries been used or has a custom code been developed for the purpose?

7.       Fig. 7 – would be better to write units on the graphs themselves.

8.       “Due to extremely small volumes, the mass of the asperities were scaled with 1000 in order to obtain longer explicit time steps and thus reach reasonable simulation times.” This sentence should be written more clearly.

9.       Fig. 11 – Please add plots of the FEM simulations depicting the pressure distribution across the domain.

10.   Use the term coefficient of friction everywhere instead of friction coefficient

11.   Fig. 21 – Is the vertical displacement of the material depicted on the figure? Please note more clearly

12.   Fig. 33 - Perhaps a log scale on the y-axis would help to depict the curves in a clearer fashion

“Although the Archard’s wear law is typically used for determination of the global wear and is much simpler to use than the advanced proposed model, it shows that it is possible to use the proposed FE model to calibrate the Archard’s wear law for a specific problem.” As stated later on, it is questionable, whether this type of Archard model calibration would yield results that are in any way meaningful for modelling a real tribological system, as in any sliding process there’d be multiple asperity contacts and resulting deformations. The authors could perhaps outline a strategy for achieving simulations that would be able to model an actual sliding contact of two rough surfaces in the future

Round 2

Reviewer 2 Report

modified report is in better stage.